# Ion Channels in the Development and Remodeling of the Aortic Valve

**DOI:** 10.3390/ijms24065860

**Published:** 2023-03-20

**Authors:** Christophe Simard, Margaux Aize, Sébastien Chaigne, Harlyne Mpweme Bangando, Romain Guinamard

**Affiliations:** 1UR 4650, Physiopathologie et Stratégies d’Imagerie du Remodelage Cardiovasculaire, GIP Cyceron, Unicaen, 14000 Caen, France; 2IHU LIRYC Electrophysiology and Heart Modeling Institute, Foundation Bordeaux, 33600 Pessac, France; 3Electrophysiology and Ablation Unit, Bordeaux University Hospital, 33600 Pessac, France

**Keywords:** ion channels, aortic valve, TRP channels, PIEZO channels, calcification

## Abstract

The role of ion channels is extensively described in the context of the electrical activity of excitable cells and in excitation-contraction coupling. They are, through this phenomenon, a key element for cardiac activity and its dysfunction. They also participate in cardiac morphological remodeling, in particular in situations of hypertrophy. Alongside this, a new field of exploration concerns the role of ion channels in valve development and remodeling. Cardiac valves are important components in the coordinated functioning of the heart by ensuring unidirectional circulation essential to the good efficiency of the cardiac pump. In this review, we will focus on the ion channels involved in both the development and/or the pathological remodeling of the aortic valve. Regarding valve development, mutations in genes encoding for several ion channels have been observed in patients suffering from malformation, including the bicuspid aortic valve. Ion channels were also reported to be involved in the morphological remodeling of the valve, characterized by the development of fibrosis and calcification of the leaflets leading to aortic stenosis. The final stage of aortic stenosis requires, until now, the replacement of the valve. Thus, understanding the role of ion channels in the progression of aortic stenosis is an essential step in designing new therapeutic approaches in order to avoid valve replacement.

## 1. Introduction

Heart valves are essential in order to provide healthy blood circulation. Indeed, they permit a unidirectional flow across heart chambers, allowing the latter to properly perform its function of the pump. Valves are subject to intense activity since they open and close approximately 3 billion times during a human’s life. Given this major role, the dysfunctions of the valves lead to strong cardiac disorders, often degenerative, which are difficult to correct other than by valve replacement, in particular, due to the poor knowledge of the molecular actors involved. The disorder comes from either an abnormal development of the valves or from their remodeling over time, induced by a wide variety of factors. This remodeling is mainly related to a stiffening of the valve leaflets, which leads to more difficult valve openings, deleterious cardiac mechanical stress and reduced oxygen supply to the body. Imperfect closure of the valve can also occur, giving way to regurgitation that reduces, in fine, the efficiency of the cardiac pump.

Among the four cardiac valves, dysfunction of the aortic valve, which is located at the junction between the left ventricle and the aorta, is particularly involved in blood circulation disorders. Indeed, this valve is subjected to important mechanical constraints, according to the strong pressure exerted at the exit of the left ventricle during systole. Two major aortic valve disorders can be observed. On the one hand, the bicuspid aortic valve is a malformation of congenital origin in which the valve has only two leaflets instead of three. The bicuspid aortic valve is observed in 1 to 2% of the population [1]. On the other hand, aortic stenosis results from the remodeling of the leaflets with the development of fibrosis and, in the most severe stages, the presence of mineralization. This stenosis has three main causes: congenital malformations, including bicuspid valve, which represent 5% of cases of the pathology and its leading cause in patients under 65 years old in Europe [1,2]; degenerative aortic stenosis, which is the most frequent form, representing 82% of cases in Europe [2]; finally, rheumatic aortic stenosis due to a bacterial infection causing inflammation and degradation of the extracellular matrix within the leaflet. This last form represents only 11% of cases in industrialized countries [2,3]. One can also note the appearance of aortic stenosis following exposure to ionizing radiations, as has been reported in some patients subjected to radiotherapy in the context of thoracic malignancies treatments [4,5]. Altogether, aortic stenosis is observed in 2% of the population over 65 years old, 2.8% in the population over 75 years old and 4% in the population over 85 years old, indicating an increase in incidence in the elderly [6,7]. Thus, it predicts a future rise in the occurrence of this disease with the enhancement of life expectancy in the world population. In this context and regarding these observations, the early detection and management of patients with aortic valve disorders seem to be essential but need further investigations.

## 2. Histological Structure of the Aortic Valve and Remodeling

The aortic valve is formed during embryonic cardiac development through a process known as valvulogenesis. This valve is normally composed of three leaflets attached to a fibrous annulus (Figure 1). Leaflets are formed with a succession of three layers of conjunctive tissue named from the aorta to the ventricle: *fibrosa*, *spongiosa* and *ventricularis*. Together, these three layers work to maintain the structural integrity and function of the aortic valve. Each layer is composed of an abundant extracellular matrix with mainly collagen fibers in the *fibrosa*, proteoglycans in the *spongiosa*, and elastin fibers in the *ventricularis* [8]. The major cells embedded in this extracellular matrix are valvular interstitial cells (VIC) which are fibroblastic cells. Leaflets are covered by a single layer of valvular endothelial cells (VEC) [9] (Figure 1). Note that during development, these endothelial cells are the origin of VIC after migration and endothelial-mesenchymal transition [10].

Pathological aortic valve remodeling can be onset by a variety of insults resulting in the disruption of the endothelial barrier, such as transient infection, mechanical stress or radiation, as observed during radiotherapy. In the context of endothelial barrier degradation, a subendothelial accumulation of lipids and invasion of the leaflets by immune cells such as monocytes and T cells was reported [11]. After infiltration in the valvular tissue, monocytes differentiate into macrophages which release inflammatory cytokines, such as tumor necrosis factor-alpha (TNFα) and transforming growth factor beta1 (TGFβ1) in the valvular extracellular matrix [12,13]. It thus onset the differentiation of VIC, which, at term, will drive to valvular fibrosis and calcification. In the healthy tissue, VIC are mostly in a quiescent phenotype representing about 95% of the total VIC population. These cells are responsible for the secretion of extracellular matrix components. Under TGFβ1 and TNFα stimulation, quiescent VIC will activate and progress to a myofibroblastic phenotype which is characterized by an intracellular expression of α-smooth muscle actin (α-SMA) and production of extracellular matrix proteins, mainly collagen, which thickens the leaflets and is responsible for fibrosis. A further step in the development of aortic stenosis is the mineralization of the valve due to the transition of VIC into osteoblast-like VIC characterized by a variety of osteoblastic markers, such as osteopontin, bone morphogenetic protein 2 (BMP2) and Runt-related transcription factor 2 (RUNX2) [9,14]. Note that VEC are also able to differentiate into osteoblasts through an endothelial-mesenchymal transition [15]. Osteoblasts will be responsible for hydroxyapatite (calcium-phosphate) crystals formation in the extracellular matrix and, thus, valvular mineralization.

Two phenomena are particularly involved in aortic valve formation as well as valve remodeling: firstly, the mechanical stress which participates in appropriate valve formation but also in the stimulation of pathological remodeling; secondly, the Ca^2+^-homeostasis with its large panel of regulatory processes [10]. Note that mechano-transduction processes and Ca^2+^-signaling closely interact in valvular remodeling [16]. These two phenomena are also the two major points in which ion channels are implicated. Ca^2+^ enters and exits the cell mainly through ion channels to modulate cellular functions. On the other hand, mechano-sensation is achieved by mechano-sensitive channels. It can be pointed out that academic research on ion channels implicated in extracellular sensing was rewarded very recently with the awarding of the 2021 Nobel prize in physiology to Ardem Patapoutian for its work on mechano-sensitive channels Piezo and to David Julius for its work on Ca^2+^-permeable Transient Receptor Potential (TRP) channels [17].

Altogether, ion channels appear as relevant targets to better understand healthy and pathological aortic valve behavior. The present review will focus on ion channels involved in aortic valve development and remodeling, including a strong implication of TRP and Piezo channels according to their participation in Ca^2+^ flux and mechano-sensation.

## 3. The Piezo and TRP Channel Families

Transient Receptor Potential (TRP) proteins are ion channels whose genes were cloned between 1989 and the early 2000s, with actually 27 identified members of the family in humans. These are classified into subfamilies named TRPA (ankyrin; TRPA1), TRPC (canonic; TRPC1, TRPC3-TRPC7), TRPM (melastatin; TRPM1-TRPM8), TRPML (mucolipin; TRPML1-TRPML3), TRPP (polycystin; TRPP1-TRPP3) and TRPV (vanilloid; TRPV1-TRPV6) [18]. These channels are non-selective cation channels, most of them harboring Ca^2+^-permeability except TRPM4 and TRPM5, which are only permeable to monovalent cations but are, however, activated by internal Ca^2+^. According to this, TRP channels participate in maintaining intracellular Ca^2+^ homeostasis and are implicated in a wide variety of Ca^2+^-dependent mechanisms, in particular in sensory processes, by transducing an external stimulus into an intracellular Ca^2+^ variation. TRP channels are widely expressed in tissues, and quite all cell types express several TRP isoforms. For reviews regarding tissues that might be implicated in valvular properties, one can note TRP channel expression in cardiac tissue [19], in vasculature [20], and in immune cells [21]. At the molecular level, TRP channels have a classical structure of homotetramers, each subunit holding six transmembrane-spanning segments. Channels from the TRP family are poorly voltage dependents. Several members, including TRPV4, were identified as mechano-sensitive channels, but it is not clear if these channels are the primary mechano-sensors or are secondary actors in mechano-sensory signaling cascades [22].

Ten years after the cloning of TRP channels, another family of non-selective cation channels was identified and characterized as a specific mechanically activated channel named Piezo, with only two members of the family (Piezo1 and Piezo2) in humans [23,24]. Piezo channels are permeable to cations with a slight preference for Ca^2+^ over Na^+^ and K^+^ [23]. Accordingly, mechanical activation of Piezo channels leads to an inward Ca^2+^ current responsible for mechanical-induced changes in cell signaling. Piezo1 is expressed in nonsensory tissues, playing a role in mechano-adaptation to fluid shear stress or cell stretching in vessels, bladder, lung skin and red blood cells [25]. Oppositely, Piezo2 acts as a major mechano-transducer in sensory tissues, being implicated in touch, proprioception and control of breathing through the Hering-Breuer reflex [25,26]. At the molecular level, Piezo has an original structure in the world of ion channels, being mastodon, since they are trimeric proteins, each subunit having 2521 and 2752 amino acids for human Piezo1 and Piezo2, respectively. While not yet determined in humans, each subunit of mice Piezo1 was shown to have 38 transmembrane-spanning segments [27,28]. The specific arrangement of the protein allows a strong mechano-transduction mechanism with a lever-like behavior that amplifies mechanical changes to modulate channel activity [25,29]. Note that, depending on studies, Piezo channels were shown to be activated by either membrane tension after cell stretching and/or fluid shear stress.

## 4. Ion Channels in Aortic Valve Development

It has been shown that several types of ion channels are involved in the development of the aortic valve. Their identification was made after observation of abnormal development of the aortic valve in patients carrying a mutation of the gene coding for one of these channels or by the use of transgenic animal models (zebrafish or mouse). These channels are described in the next section and summarized in Table 1.

### 4.1. Piezo1 Channel in Aortic Valve Development

A demonstration of the implication of ion channels in human aortic valve development recently came with the identification of mutations of the *Piezo1* gene in patients harboring bicuspid aortic valve [30]. They correspond in the protein to p.S217L, p.2022H, and p.K2502R amino acids substitutions and appear to be dominant negative isoforms. The effect of the mutations on channel properties were evaluated after expression in the HEK-293 cell line. It revealed that mutations are loss of function variants since they result in a decreased current amplitude compared to wild-type (WT) Piezo1, arising from a deleterious effect on channel functions rather than protein expression [30]. The S217L mutant behavior was further studied by another group which observed reduced stability of the protein and a higher turnover rate than WT with increased ubiquitination [36]. The implication of Piezo1 in aortic valve formation was also evaluated in an experimental model using embryonic zebrafish. The aortic valve is formed from the outflow tract, which is a transient structure whose development is strongly influenced by hemodynamic forces. Knockdown of the *piezo1* gene in zebrafish leads to a dysmorphic and misshapen valve with enlargement, compared to WT animals, which confirms the critical role of the channel in aortic valve development [30]. Note that the implication of Piezo1 was also previously observed for lymphatic valve formation in mice [37].

Even if the mechanisms by which Piezo1 modulates valve development are not yet fully determined, several possible pathways were identified. Valve formation is a complex process that comes from an endothelial-mesenchymal transition and is dependent on the biomechanical factors arising from blood flow-induced shear stress. This process is mediated by BMP2 and Notch signaling pathways [10,38,39]. Inhibition of Notch1 signaling during zebrafish development interrupts valve development by decreasing the endothelial-mesenchymal transition [40]. In addition, *Notch1* mutations were identified in patients with congenital aortic bicuspid valves [41]. Upstream of Notch1, ADAM10 (A disintegrin and metalloproteinase 10) was shown to participate in embryogenic cardiovascular development through Notch1 stimulation [42,43,44]. ADAM10 is a Ca^2+^-regulated transmembrane sheddase that mediates S2 Notch cleavage. It is in this context that the Piezo1 channel might be implicated. Indeed, it has been shown to regulate Notch1 signaling by activating ADAM10 sheddase under shear stress [45].

Besides this, the hippo/YAP1 signaling pathway known to control the transcription of target genes was also shown to be involved in zebrafish Piezo1-dependent aortic valve development [46]. Valve formation was delayed by *Piezo1* knock-down, and the authors observed thicker leaflets than in WT animals [31]. The *Piezo1* knock-out embryo exhibits a reduced expression of YAP1 in the two layers which compose the outflow tract, corresponding to the aortic valve, which is composed of endothelial cells and smooth muscle cells. *Piezo1* knock-out embryos also exhibit an increase in the mechano-sensitive transcription factor klf2a [31,46]. This factor is known as a flow-responsive gene in zebrafish as well as in vertebrates [33]. Collectively these data suggest that Piezo1 appears to modulate outflow tract development by inhibiting klf2a in the endothelium and stimulating Yap1 in the endothelium and smooth muscle [46].

### 4.2. TRP Channels in Aortic Valve Development

A recent investigation has shown that the knock-down of the *TRP* genes *Trpp2* and *Trpv4* both result in valvular zebrafish development impairments [31]. *Trpp2* knock-out fish exhibit delayed valve formation, while *Trpv4* knock-out fish exhibit thickening of the valve. These two channels were previously shown to modulate the mechano-sensitive klf2a expression and Ca^2+^ homeostasis in zebrafish [33]. The authors first observed that intracardiac flow stimulated klf2a expression in the endocardium, which was correlated with an increase in intracellular Ca^2+^ levels in the endocardium. These two phenomena were prevented by *Trpp2* as well as *Trpv4* gene disruption. Altogether, it indicates that these believed mechano-sensitive channels might influence aortic valve development by producing a Ca^2+^-inward current which onsets the Ca^2+^-activated intracellular signaling cascade leading to klf2a expression, thus influencing development. However, a new report may weaken this interpretation. Indeed, while TRPV4 was initially described to be mechano-sensitive [47], this concept was stressed by a recent study reevaluating the stretch sensitivity of a variety of TRP isoforms after expression in the HEK-293 cell line [22]. In this specific cell line, TRPV4 did not appear to respond to membrane stretch. It is thus believed that this channel would not be the primary mechano-sensor but could be activated by mechano-sensing partners proteins.

TRPM4 might also be a candidate for bicuspid aortic valve susceptibility. *Trpm4* knock-out C57Bl/6 mice are known to develop cardiac hypertrophy with perturbation of electrical activity [48,49]. In a recent study evaluating aortic valve morphology by transversal slicing, we reported bicuspid valves in 17% (2/12) of *Trpm4* knock-out animals, while all WT mice tested in this study exhibited tricuspid valves (19/19) [32]. According to the small number of animals used in this specific study, the relation between phenotype and genotype remains to be confirmed. However, an extensive review of the literature reported that in overall studies using this mouse strain, only 0.3% (2/366) WT animals exhibited bicuspid aortic valves [50]. This is significantly lower than the occurrence in *Trpm4* knock-out animals. The mechanism linking *Trpm4* disruption to valvular abnormalities is unknown but might be related to intracellular Ca^2+^ signaling since this channel is Ca^2+^-dependent. To our knowledge, no data are available in humans, even if several *TRPM4* mutations were reported in patients with cardiac electrical perturbations [51,52]. It could be valuable to investigate aortic valve morphology in families with loose functions *TRPM4* mutations, in particular for the K914X mutation, which leads to a non-functional protein [53] and, thus, might resemble the effect of knocking out of the gene in mice.

### 4.3. Other Ion Channels in Aortic Valve Development

Two other channel types were associated with bicuspid aortic valves in humans. First, the inward-rectifying potassium channel Kir2.1 is encoded by the *KCNJ2* gene. A heterozygous missense mutation R67W was identified in a family with ventricular arrhythmia and periodic paralysis. 4/41 mutation carriers had a bicuspid aortic valve [34]. Heterologous expression of the mutant in the tsA201 cell line and Xenopus oocyte indicated a total loss of channel function. Co-injection of WT and mutant in Xenopus oocyte also resulted in a loss of function, indicating a dominant negative effect of the mutant residue [34]. To our knowledge, no other mutations of this gene were reported to be associated with valvular malformation, and no data are available on animal models.

The intracellular Ca^2+^-activated chloride channel anoctamin-5 from the TMEM16 protein family and encoded by the *ANO5* gene might also be responsible for aortic valve malformation. Mutations of this gene are responsible for muscular dystrophy, but a study also revealed cardiac disturbances in patients with bi-allelic mutations. 1/10 of these patients exhibited a bicuspid aortic valve, and 4/10 had a thickened aortic valve but without aortic stenosis or insufficiency [35]. Note that quite all these patients hold the c.191dup mutation leading to a p.N64Kfs insertion frameshift, which is the most common pathogenic variant in *ANO5*-related muscle disease [35,54]. Once again, the mechanism linking the mutation to valvular abnormalities is unknown but might be related to intracellular Ca^2+^ signaling since this channel is Ca^2+^-dependent.

## 5. Ion Channels in Aortic Valve Remodeling

Several types of ion channels have been shown to participate in aortic valve remodeling. Some of them have been identified through experiments conducted in vivo on animal models of aortic stenosis, whereas others have been identified using in vitro models of isolated aortic valve cells. Note that the strongest data is for the VIC, while the data for the other cell types remain weak. The main channels involved in aortic valve remodeling are summarized in Table 2.

VIC are the valvular cell type that received the most attention due to the fact that they are the most prevalent cell type in the valve and are responsible for maintaining valve integrity and stiffness by secreting appropriate extracellular matrix components. On the other hand, VIC are also implicated in valvular pathological remodeling following their activation and transition to myofibroblastic or osteoblastic phenotypes leading to fibrosis and calcification, respectively [9]. VIC are under the control of a variety of factors (TGFβ1, TNFα...) secreted by other cell types, such as VEC and immune cells (Figure 2). They are also under the control of mechanical factors according to the leaflets’ deformation at each heartbeat. Indeed, such mechanical stress is known to induce porcine VIC myofibroblast activation [59]. According to this, VIC are influenced and controlled by the extracellular matrix composition of the leaflets.

A variety of ion channels were identified in VIC, and most of them might be involved in Ca^2+^-signaling and/or mechanical sensitivity. Even if their implication in VIC function and remodeling is not fully understood, several recent studies confirmed the implication of few ion channels in these processes. Note that several of them were already implicated in valve formation, such as TRP channels, as described in previous sections.

### 5.1. TRP Channels in VIC and Aortic Valve Remodeling

At least three TRP isoforms are expressed in human VIC, and those were searched for their mechano-sensitivity. These are TRPC6, TRPM4 and TRPV4 isoforms [55]. Their protein expression was evaluated by western blot on isolated human VIC cultured in condition to drive their differentiation into either fibroblastic, myofibroblastic or osteoblastic phenotypes. It appeared that TRPC6 and TRPV4 expression levels were significantly higher in osteoblastic VIC compared to the other phenotypes, while TRPM4 was also detected at a higher level in osteoblastic VIC even if not reaching significance, compared to other phenotypes [55]. Immunohistochemical labeling of those channels on leaflets revealed that the three channels are strongly expressed by VIC originating from patients with calcified aortic valves but not from patients with non-calcified ones [55]. To our knowledge, the recording of functional currents driven by those channels on VIC has not yet been published. However, those three channels are known to be functionally expressed by cardiac fibroblasts and, thus, might also be functional in VIC according to their resemblance with those cells [19,60,61,62]. Comparison with other models suggests that these TRP channels might be implicated in VIC remodeling, as described below.

In a model of primary culture of cardiac fibroblasts from rats, TRPC6 overexpression was shown to induce an increase in α-SMA stress fibers and collagen I synthesis, indicating a transdifferentiation into myofibroblast [60]. Cardiac fibroblast stimulation by TGFβ induces an increase in TRPC6 mRNA and protein levels, which occurs through the p38-MAPK pathway. Interestingly, this stimulation by TGFβ-p38 signaling induces a TRPC6-dependant store-operated Ca^2+^-entry promoting myofibroblast transformation after activation of the calcineurin-NFAT (nuclear factor of activated T cell) pathway [60]. Since VIC are fibroblastic cells, it is feasible that TRPC6 participates in their transition to myofibroblasts through the same pathways.

Similarly, TRPM4 was also shown to participate in cardiac fibroblast transdifferentiation. Indeed, in a model of mouse atrial cardiac fibroblasts in a culture that undergoes growth and transdifferentiation into myofibroblasts with an increase in α-SMA levels, it was observed that these phenomena were reduced by *Trpm4* gene disruption [61]. Similar results were also obtained with pharmacological inhibition of TRPM4 by 9-phenanthrol in human atrial fibroblasts in culture [61]. Furthermore, TRPM4 was also shown to be implicated in aortic valve remodeling in an in vivo model of radiation-induced valve remodeling [32]. In that study, it was observed that irradiation of the valve produces a functional remodeling of the valve after 5 months with an increase of maximal jet velocity in WT mice which is reduced in *Trpm4* knock-out animals. At the morphological and histological levels, irradiation induces leaflets thickening and fibrosis, which are less pronounced in *Trpm4* knock-out mice than in WT [32]. Altogether it indicates that TRPM4 might be implicated in VIC transdifferentiation leading to valvular fibrosis.

The last candidate, TRPV4, might be Implicated in the mechanical sensitivity of VIC. Indeed, stretch-induced increase in collagen I synthesis by human VIC in culture is reduced by the pharmacological TRPV4 inhibitor RN 9398 [55]. More generally, TRPV4 was shown to be implicated in several models of fibroblast mechano-sensitivity [63]. Its role in valvular stiffness-induced myofibroblastic activation was also investigated in a model of isolated porcine VIC in culture [56]. To evaluate the influence of mechanical stress, VIC were cultured in several hydrogel biomaterials with different stiffness. Data indicate that a stiff microenvironment induces a myofibroblast activation which is prevented by TRPV4 inhibition using GSK219 [56]. This regulation was shown to also involve the Yes-activated protein (YAP) pathway as a downstream target for TRPV4 activity [56]. Finally, the mechanical-induced myofibroblasts activation was observed to be under the control of the TRPV4-dependent intracellular Ca^2+^ variations in these porcine VIC in culture [56,57]. The implication of TRPV4 in interactions between mechano-sensing and Ca^2+^-signaling, leading to extracellular matrix remodeling, was extensively reviewed recently [16]. It indicates a cross-talk between TRPV4 and extracellular remodeling. On the one hand, TRPV4 is dependent on tissue stiffness. On the other hand, extracellular matrix composition is dependent on TRPV4 activation and, thus, Ca^2+^ signaling, in particular with the activation of myofibroblast [16].

As detailed above, the direct mechano-sensitivity of TRP channels is under debate since a recent reevaluation of this property showed no direct effect on TRP channels overexpressed in HEK293 cells [22]. If so, TRP activation would necessitate an intermediate protein to sense mechanical stimuli. Piezo1 appears as an attractive candidate for that role, even if, to our knowledge, its expression was not reported yet in VIC. An example of this Piezo/TRP interaction could be provided for the TRPM4 channel. Indeed, TRPM4 was shown to participate in pressure overload-induced cardiac hypertrophy in mice [64]. However, in a further study using the same model, it was shown that Piezo1 was the initial sensor of mechanical stress in mice cardiomyocytes, which allows Ca^2+^-entry leading to TRPM4 activation since this latter is a Ca^2+^-activated channel [65]. This finally leads to the activation of the Ca^2+^/calmodulin-dependent kinase II (CaMKII), inducing the hypertrophic pathway [64,65]. The implication of Ca^2+^-influx through Piezo1 to activate channels most probably does not apply for TRPV4 since this latter is not Ca^2+^-dependent. Thus, other proteins would allow the TRPV4 mechano-sensitivity, for example, interactions with cytoskeletal or external proteins. β1-integrin might be a good candidate since it was shown in bovine capillary endothelial cells that TRPV4 is activated by mechanical forces through β1-integrin cytoplasmic tail binding protein CD98 [63,66]. While not demonstrated in VIC, this process is, however, plausible since β1-integrin was detected in porcine VIC in culture [67]. Finally, several additional TRP isoforms (TRPA1, TRPC3, TRPM2, TRPM7, TRPV1, TRPV3) were reported to be expressed in cardiac fibroblasts and, thus, appear to be interesting to be searched for in VIC [68].

### 5.2. Voltage-Dependent Ca^2+^ Channels in VIC and Aortic Valve Remodeling

Voltage-dependent Ca^2+^ channels (VDCa) also appear as feasible pathways for Ca^2+^ entry in VIC and, thus, may participate in remodeling. A genome-wide association study revealed an up-regulation of the *CACNA1* gene in aortic valves from patients with calcific aortic valve disease [14]. *CACNA1* encodes for the α-1C subunit of the Cav 1.2 VDCa. It was later confirmed, at the protein level, an increase in Cav 1.2 channel in valves from patients with calcific aortic valve disease compared to control patients [58]. The casualty of Ca_v_ 1.2 overexpression in the development of valvular calcification was then tested in a transgenic mouse model with a specific overexpression of the channel in the aortic valve tissue. It results in calcified lesions [58], in accordance with the calcified phenotype observed in patients. Moreover, knock-in mouse transfection with the gain-of-function Ca_v_ 1.2 mutant G406R, specifically in the aortic valve, exacerbates this phenotype, thus highlighting the implication of the channel in the development of aortic valve calcification. An increase in RUNX2 and α-SMA was also detected by immunohistochemical labeling. Interestingly, in vivo treatment of the mice with the Ca^2+^ channel blocker verapamil was able to prevent valve calcification [58].

At the cell level, transfection of mouse VIC in culture with the *Ca_v_ 1.2* gene induced an increase of α-SMA, which, once again, was prevented by verapamil, indicating that the channel participates in the induction of the myofibroblastic VIC transition [58]. This was correlated in those cells to an increase in Ca^2+^-signaling genes expression, the highest being the transcription factor activated by the Ca^2+^-sensitive phosphatase calcineurin *Nfatc2* [58], which is known to be also upregulated in calcified aortic valve from patients if compared to non-calcified valves [69].

The implication of Ca^2+^-influx through a VDCa channel in non-excitable cells, such as VIC, might be questionable since their role is mostly known in excitable cells with strong variations in membrane potential. However, one can hypothesize VDCa activation following cell depolarization by the inward cationic current. TRP channels, as well as Piezo1, are good candidates to induce such depolarization. Reciprocally, it is conceivable that Ca^2+^ entry through VDCa modulates Ca^2+^-dependent channels, such as TRPM4. Such activation may have complex effects. On the one hand, TRPM4 activation would increase cell depolarization and, thus, VDCa activation but on the other hand, TRPM4-induced cell depolarization would reduce the inward driving force for Ca^2+^, as was shown in pancreatic acinar cells [70].

### 5.3. K^+^-Channels in VIC and Aortic Valve Remodeling

To date, very little is known about K^+^ channels in VIC. A recent study evaluated the role of mechanosensitive K^+^ channels in human VIC in culture. By electrophysiological recording, the authors detected a mechano-sensitive K^+^-selective current in only 5% of VIC, while a mechano-sensitive non-selective cation current was detected in quite all cells (92%) [55]. The K^+^-selective current might be supported by TREK-1 and Kir 6.1 channels since those were detected at the protein level by a western blot on human VIC. Note that Kir 6.1 is known to be the four pore-forming inward rectifier subunits of the ATP-dependent K^+^ channel (K_ATP_). Interestingly, protein expressions of TREK-1 and Kir 6.1 decreased when VIC in culture was driven to differentiate into myofibroblastic as well as osteoblastic phenotypes. Additionally, pharmacological inhibition of TREK-1 by spadin had no effect on collagen I synthesis by VIC [55]. Altogether, it indicates that K^+^ channels, at least TREK-1 and Kir 6.1, do not appear as major determinants in VIC remodeling.

### 5.4. Ion Channels in VEC and Aortic Valve Remodeling

Valvular leaflets are covered by a monolayer of endothelial cells, and a variety of ion channels are known to be expressed in such cells [71]. However, depending on their location, endothelial cells exhibit a large phenotypic heterogeneity [72], and little is known specifically for aortic VEC, which exhibit different transcriptional profiles than vascular endothelial cells [73]. According to this, we will not provide an extensive review of ion channels in endothelial cells but will only report a few data which might be relevant for aortic VEC. An important point for their physiological and pathological implication is the effect of shear stress, given that VEC lining the aortic valve are submitted to strong flux variations at each systole. Note that such flux is different between both sides of the valve. The ventricular side will sense a unidirectional laminar flux, while the aortic side will sense an oscillatory flux [74]. Consequently, the side-specific organization of the VEC will appear with differences in cell alignment and components in the actin cytoskeleton (Figure 1) [75]. VEC express ion channels allowing them to detect shear stress and transmits signals to VIC, thus influencing valvular remodeling [74,76].

As shown for VIC, TRP, as well as Piezo1 channels, might be actors of those phenomena in VEC. Immunohistochemical staining on slices from human aortic valve leaflets revealed a low protein expression of TRPC6, TRPM4 and TRPV4 in endothelial cells from each side of the leaflets [55]. On the other hand, Piezo1 was described as responsible for shear-stress evoked Ca^2+^-influx in human and mouse endothelial cells in culture, but it remains to be evaluated in VEC [77].

Kir 6.1 might be implicated in VEC mechano-sensation. Pinacidil, which activates the ATP-dependent K^+^ channels, was shown to increase cytoplasmic Ca^2+^ in mouse aortic VEC [78]. This effect was totally absent in cells from a mouse with specific endothelial knockdown of *Kir 6.1,* which indicates the implication of this channel in Ca^2+^-signaling in VEC [78]. One can hypothesize that activation of an outward K^+^ current would increase the driving force for Ca^2+^ entry. Note that a slight Kir 6.1 immunolabelling was detected on VEC from human aortic leaflets, which leads to the hypothesis that the channel might also be involved in Ca^2+^-signaling in human VEC [55].

### 5.5. Ion Channels from Immune Cells and Their Implication in Valvular Remodeling

Inflammation is one of the first steps of aortic valve remodeling. An endothelial injury that can be caused by mechanical stress leads to the infiltration of lipids, such as apolipoproteins, within the valvular tissue [79]. In association with endothelial cell injuries, it produces an inflammatory process leading to the attraction of immune cells, such as macrophages and T lymphocytes [13]. Such cells will produce a large variety of factors that influence VIC remodeling and, thus, valve fibrosis and calcification. Among them are TNFα, TGFβ1 and vascular endothelial growth factor (VEGF), which effects on VIC were described in the previous sections [79]. Note that subsequent leaflet stiffness produces additional mechanical stress leading to a vicious cycle in valvular remodeling [11]. According to these phenomena, ion channels from immune cells will indirectly participate in valvular remodeling. While extensive literature is available about ion channels in immune cells, little is known specifically about these channels in the context of valvular remodeling.

Macrophages, with their precursor monocytes, represent the major immune cell type in affected valves [13]. While not specifically demonstrated in the context of macrophages invading the valve, TRP, as well as Piezo1 channels, might be involved in the role of macrophages and T lymphocytes in aortic valve remodeling. Once again, the control by those channels of intracellular Ca^2+^ homeostasis in immune cells might be a hub in the phenomenon. Immune cells, especially macrophages and T cells, are known to express a large variety of TRP channels [21]. For example, the Ca^2+^-activated channel TRPM4 was shown to influence Ca^2+^ homeostasis in immune cells, including T cells [80,81] and macrophages [82]. TRPV4 was also shown to influence Ca^2+^-homeostasis in macrophages, leading to various secretions of immune factors, such as TGFβ, TNFα and IL-1β [83].

Like a third accomplice, Piezo1 could invite itself into the process of macrophage secretions leading to valve remodeling. Indeed, Piezo1 is expressed in macrophages and participates in the sensing of microenvironmental stiffness as well as shear stress [84,85]. Interestingly, it was shown that the Ca^2+^-permeable mechano-sensitive channel involved in monocyte shear-stress activation is Piezo1 but not TRPV4 [85]. As indicated previously, mechanical stress is one of the inductors of valve remodeling, but mechanical stress also may result from valve remodeling, thus forming a vicious circle. A nice study demonstrated the implication of Piezo1 in this circle by comparing human monocytes activation in patients with aortic stenosis, corresponding to a high mechanical stress, and in the same patients one month after transcatheter aortic valve implantation (TAVI), corresponding to a low mechanical stress environment [85]. The authors observed a reduced adhesive capability of monocytes after TAVI, as well as reduced levels of inflammatory markers, such as IL6. In-vitro data with pharmacological and biomolecular approaches demonstrated that Piezo1 was the mechano-sensor in this phenomenon [85].

## 6. Conclusions and Perspectives

The previous paragraphs show the contribution of a number of ion channels in the remodeling of the aortic valve, as well as the involvement of at least three cell types. Those actors are schematized in Figure 2, which indicates the known elements but remains with many unclear areas.

Studies reviewed above highlight crosstalk between stiffness, mechano-transduction and valvular remodeling, with Ca^2+^-signaling being a driving belt in that system. Ion channels are implicated in these phenomena at several levels: first serving as mechano-sensors (Piezo1, TRPV4), secondly allowing Ca^2+^ flux through the cell membrane (TRPC6, TRPV4, Piezo1, VDCa), and finally as Ca^2+^ flux regulators (TRPM4, Kir6.1). Ca^2+^-signaling modulators induce changes in gene expression and, thus, valvular remodeling. Even if these channels were shown to be implicated, we are far from having a clear view of the contribution of each channel and, even more, the sequence of events. The continuation of in-vitro studies will make it possible to describe this sequence for each of the three cell types involved in the phenomenon, which are macrophages, VEC, and VIC. Note that the development of 3D models of culture with appropriate hydrogels appears as a valuable approach [76]. On the other hand, understanding the in vivo phenomenon that involves at least these three cell types will be much more complicated. As an intermediate step, the experiments carried out with co-cultures [76] seem to be a good alternative and already provide some relevant elements.

Although some studies have been carried out with transgenic animals disrupted for a specific type of ion channels in zebrafish [30,31], as well as in mice [32,78], it is difficult to understand the involvement of these channels because they are often expressed in several cell types, including outside of the valve. Thus, recourse to cell-specific genetic constructs will undoubtedly be necessary.

Another pitfall comes from the difficulty of using in vivo pharmacological approaches, given the uncertainties about the specificity of the compounds used to modulate channel activity. However, several molecules targeting a specific ion channel have already been used for in vivo studies in different pathological contexts. Let site for examples in mouse or rat: yoda1 for Piezo1 activation [86]; GSK2193874 for TRPV4 inhibition [87]; 9-phenanthrol for TRPM4 inhibition [88]; verapamil for VDCa inhibition [58]. It, therefore, seems conceivable to test their effects on the remodeling of the aortic valve in vivo. The combination of this pharmacological approach with molecular approaches will probably allow significant progress.

Increasing the knowledge of the actors involved in the development of aortic stenosis will make it possible to move forward in order to propose innovative and adequate therapies, while currently, only valve replacement is proposed for the advanced stages of the disease. Even if the ion channels are not necessarily the cause of the problem, their participation in the processes leading to stenosis makes them interesting therapeutic targets to slow down the deleterious remodeling of the valve. Therapeutic approaches to treat ion channel de-pendent disorders often target the symptoms rather than the channel itself. However, a number of approaches are being developed to specifically target channels in a variety of diseases, the most common being the pharmacological approach [89]. The identification of the ion channels involved in aortic stenosis, coupled with the development of approaches allowing the modulation of these channels, should make it possible to see the emergence of effective targeted therapies.

## Figures and Tables

**Figure 1 ijms-24-05860-f001:**
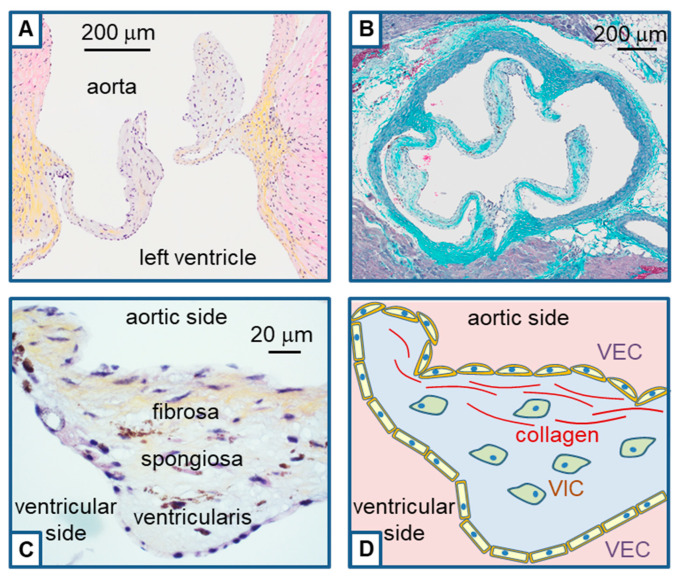
Structure of the aortic valve. (**A**): Histological slice performed in the longitudinal axis (3-μm-thickness) of the aortic valve of a mouse. Aorta is on the upper side and the left ventricle in the lower side. The slice was stained with hematoxylin, eosin and saffron (HES) to reveal cell nuclei (blue), cytoplasm (pink), and collagen fibers (yellow). (**B**): Histological slice performed in the transversal axis (3-μm-thickness) of the aortic valve of a mouse, allowing the observation of the three leaflets. The slice was stained with Masson trichrome revealing collagen fibers in cyan. (**C**): Magnification of a slice performed in the longitudinal axis (3-μm-thickness) of the aortic valve of a mouse at the level of a leaflet (HES staining). Collagen fibers (yellow staining) appear in the *fibrosa*. Note the difference in endothelium morphology between the aortic and ventricular sides. (**D**): Outline of the slice presented in C to highlight valvular endothelial cells (VEC) lining the leaflet and valvular interstitial cells (VIC) within the leaflet.

**Figure 2 ijms-24-05860-f002:**
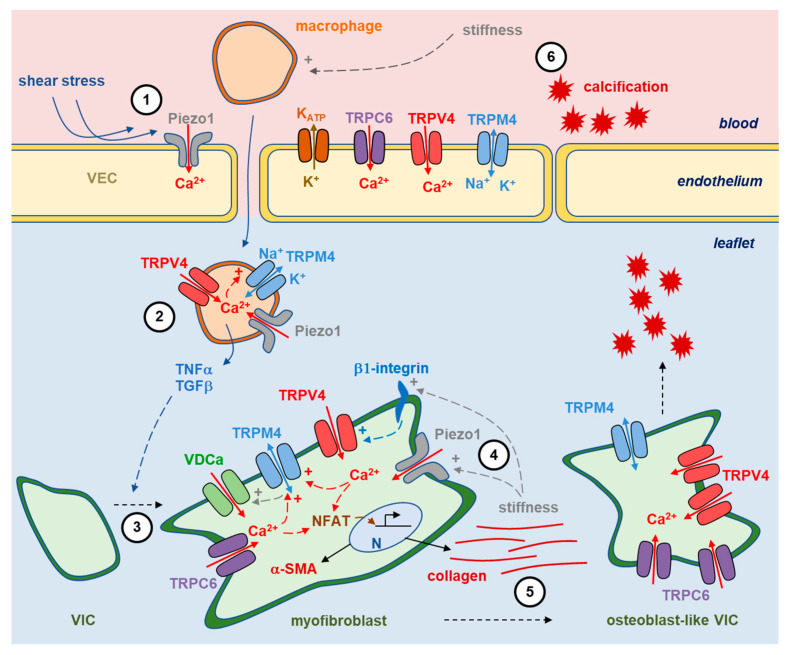
Ion channels in aortic valve remodeling. Cartoon highlighting the three main cell types which participate in aortic valve remodeling with main ion channels involved in the phenomenon. Note that several actors were demonstrated specifically in aortic valves, while others remain to be confirmed, as indicated in the main text. **1**—Valvular insults, such as mechanical stress, result in endothelium disruption. Such stimuli might be sensed by a variety of mechano-sensitive ion channels at the membrane of the valvular endothelial cells (VEC), Piezo1 being a valuable candidate. **2**—It is followed by an invasion of the leaflet by immune cells, such as macrophages. Those cells are endowed with Ca^2+^-permeable (Piezo1, TRPV4) or Ca^2+^-regulated (TRPM4) channels which influence the secretion of factors such as TNFα and TGFβ. **3**—Those factors promote the transition of quiescent valvular interstitial cells (VIC) to a myofibroblastic phenotype leading to the synthesis of α-SMA within the cell and collagen in the extracellular matrix. The phenomenon is under the control of intracellular Ca^2+^ signaling involving TRP channels (TRPC6, TRPM4, TRPV4) and voltage-gated Ca^2+^ channels (VDCa). **4**—Leaflet stiffening exacerbates such expression, probably according to its effect on the mechano-sensitive channel Piezo1 and the TRPV4 channel through interaction with β1-integrin. **5**—In the most advanced stages, VIC differentiate into osteoblast-like cells with an enhanced expression of TRP channels and calcium-phosphate deposition into and at the external surface of the leaflet. **6**—It results in additional stiffening of the leaflet, stimulating further invasion by macrophages.

**Table 1 ijms-24-05860-t001:** Ion channels are involved in aortic valve development.

Channel	Selectivity	Regulation	Effect on Aortic Valve Development	Reference
Piezo1	Ca^2+^ > Na^+^, K^+^	Mechano-sensitive	- Loss of function mutations in human: bicuspid valve - Knock-down in zebrafish: enlarged and misshapen valve with thickening	[30] [30,31]
TRPM4	Na^+^, K^+^	Ca^2+^-activated	Knock-out in mice: increase in bicuspid valve occurrence?	[32]
TRPP2	Ca^2+^ > Na^+^, K^+^	Mechano-sensitive?	Knock-out in zebrafish: delayed valve formation	[31,33]
TRPV4	Ca^2+^ > Na^+^, K^+^	Mechano-sensitive?	Knock-out in zebrafish: thickening of the valve	[31]
Kir2.1 (*KCNJ2* gene)	K^+^	Inward rectifier	R67W loss of function mutation in human: increase in bicuspid valve occurrence	[34]
TMEM16E (*ANO5* gene)	Cl^−^	Ca^2+^-activated	N64Kfs mutations in human: thickened aortic valve	[35]

**Table 2 ijms-24-05860-t002:** Ion channels involved in aortic valve remodeling. DAG: Diacylglycerol; SOCE: store-operated Ca^2+^ entry; VDCa: voltage-dependent Ca^2+^ channel.

Channel	Selectivity	Regulation	Effect on Aortic Valve Remodeling	Reference
TRPC6	Ca^2+^ > Na^+^, K^+^	DAG-activated, SOCE	High expression in human osteoblastic VIC (in vivo)	[55]
TRPM4	Na^+^, K^+^	Ca^2+^-activated	Promotes radiation-induced valvular fibrosis and thickening in mice (in vivo)	[32]
TRPV4	Ca^2+^ > Na^+^, K^+^	Mechano-sensitive?	- High expression in human osteoblastic VIC (in vivo) - Promotes collagen 1 synthesis by human VIC (in vitro) - Promotes activation of myofibroblastic porcine VIC (in vitro)	[55] [55] [56,57]
CaV_1.2_ (*CACNA1* gene)	Ca^2+^	Voltage-gated(VDCa)	- Upregulated in calcified human aortic valve (in vivo) - Specic overexpression in mouse aortic valve: calcified lesions (in vivo) - Overexpression in mouse VIC: myofibroblastic transition (in vitro)	[14,58] [58] [58]

## Data Availability

Not applicable.

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
