# Peer review of "Ion Channels in the Development and Remodeling of the Aortic Valve"

_ijms, 2023, doi:10.3390/ijms24065860_

Round 1

Reviewer 1 Report

This manuscript provides an overview of the role of ion channels in aortic valve development and pathological remodeling, emphasizing the interplay between mechanical stress and Ca2+ homeostasis. The authors delve into the molecular mechanisms of the Piezo and TRP channel families, as well as other ion channels in VICs, VECs, and immune cells (macrophages and T lymphocytes). The review concludes that Ca2+-signaling and ion channels are involved in aortic valve stiffness, mechano-transduction, and valvular remodeling, with in vitro studies and co-cultures offering valuable insights. The manuscript is well-structured, comprehensive, and relevant, with recent and pertinent references supporting clear and coherent statements and conclusions. The figures and schemes are thoughtfully designed, effectively presenting data in a visually intuitive manner.

Reviewer 2 Report

Overall well written and comprehensive review that illustrates the in depth work of the authors. This literature review article provides a state-of-the-art comprehensive summary of ion channels associated with the development of aortic valve stenosis. While the topic is not original, it is highly relevant to cardiac physiology and to aortic stenosis, which is the most common valvular disease. Compared to other published studies, it provides an aggregate of current literature on ion channels that are associated with aortic valve stenosis.

In order to increase readership (especially among those who practice cardiology), I would recommend adding more information about medications that are being developed to target ion channel anomalies.

The conclusion provides a practical summary of the literature review and likely future targets of research. The references are appropriate and extensive. The illustrations provided in the article are useful, but I would recommend adding a table that summarizes all of the channelopathies associated with aortic valve stenosis.

I am awaiting the application of these ion channel findings in pharmacotherapy.

Reviewer 3 Report

This review details ion channels and it role in aortic development and remodelling. 

This article has managed to address the key role of ion channels in aortic valve. The references have also been utilised very well.

Comments:

I would suggest adding a figure or table to sum up this part at section 4 (Ion

channels in aortic valve development )
